# Neuroimmunoendocrinology of SARS-CoV-2 Infection

**DOI:** 10.3390/biomedicines10112855

**Published:** 2022-11-08

**Authors:** Giuseppe Bellastella, Paolo Cirillo, Carla Carbone, Lorenzo Scappaticcio, Antonietta Maio, Graziella Botta, Maria Tomasuolo, Miriam Longo, Alessandro Pontillo, Antonio Bellastella, Katherine Esposito, Annamaria De Bellis

**Affiliations:** 1Unit of Endocrinology and Metabolic Diseases, University of Campania “Luigi Vanvitelli”, 80138 Naples, Italy; 2Department of Advanced Medical and Surgical Sciences, University of Campania Luigi Vanvitelli, 80138 Naples, Italy; 3University of Campania ‘‘Luigi Vanvitelli’’, 80138 Naples, Italy

**Keywords:** COVID-19, hypothalamus, pituitary gland, ACTH, cortisol, pituitary autoimmunity

## Abstract

This review is aimed at illustrating and discussing the neuroimmune endocrinological aspects of the SARS-CoV-2 infection in light of the studies on this topic that have so far appeared in the literature. The most characteristic findings and pending controversies were derived by PubMed and Scopus databases. We included original and observational studies, reviews, meta-analysis, and case reports. The entry of the coronavirus into susceptible cells is allowed by the interaction with an ecto-enzyme located on human cells, the angiotensin-converting enzyme 2 (ACE2). SARS-CoV-2 also targets the central nervous system (CNS), including hypothalamic-pituitary structures, as their tissues express ACE2, and ACE2 mRNA expression in hypothalamus and pituitary gland cells has been confirmed in an autoptic study on patients who died of COVID 19. SARS-CoV-2 infection may cause central endocrine disorders in acute phase and in post-COVID period, particularly due to the effects of this virus at CNS level involving the hypothalamic-pituitary axis. The aggression to the hypothalamus-pituitary region may also elicit an autoimmune process involving this axis, responsible consequently for functional disorders of the satellite glands. Adrenal, thyroid and gonadal dysfunctions, as well as pituitary alterations involving GH and prolactin secretions, have so far been reported. However, the extent to which COVID-19 contributes to short- and long-term effects of infection to the endocrine system is currently being discussed and deserves further detailed research.

## 1. Introduction

Since the first appearance of the novel coronavirus in Wuhan, China, in late December 2019, the outbreak has spread rapidly and globally, with cases reported in almost all countries, so that, as of 19 May 2022, it has infected over 520 million people and has claimed over 6 million lives [1]. The SARS-CoV-2 binds to the cells of the respiratory mucosa (nose, mouth, etc.) by interacting with an ectoenzyme located on human cells, the angiotensin-converting enzyme 2 (ACE2), to gain entry into the cells. ACE2 is located in large numbers on the cells of the upper respiratory tract epithelium, as well as those of the intestinal tract, upper and lower, heart and kidney. The spike protein present on the surface of the coronavirus binds well to the cell ACE2 receptor and allows the virus to fuse with the cell and release its genome, made up of an RNA chain, into the cells [2]. In conclusion, the entry of the coronavirus into susceptible cells is a complex process that requires the concerted action of the binding to the receptor and the proteolytic processing of protein S to promote the fusion between virus and cell. One of the cellular proteases involved in the entry of CoV-2 into host cells is the transmembrane serine protease 2 (TMPRSS2) [3]. In endocrine organs, SARS-CoV-2 infection may worsen existing diseases or cause new abnormalities [4]; such endocrine diseases may specularly worsen the prognosis of COVID-19 [5,6,7]. Endocrine dysfunctions are reported in several gland secretions involving adrenal, thyroid, gonads, and are associated with anxiety, stress, and depression [8]. This review illustrates and discusses the neuroimmune endocrinological aspects of SARS-CoV-2 infection in light of the studies on this topic that have so far appeared in the literature.

## 2. Methods

We revised English-published papers underlying the mentioned aim, derived from PubMed and Scopus databases and published from 2002, the year of the appearance of the first SARS-CoV-1 infection, to 31 May 2022. Moreover, we also revised papers published from 1975 to the present, focusing on some historical aspects of endocrine autoimmunity. We included original and observational studies, reviews, meta-analysis, and case reports, following the combination of the keywords “SARS-CoV, COVID-19, hypothalamus-pituitary autoimmunity, SARS-CoV and endocrine dysfunctions”.

## 3. Involvement of Central Nervous System and Hypothalamic-Pituitary Axis in SARS-CoV-2 Infection

The SARS-CoV-2 also targets the central nervous system (CNS). Therefore, after the first descriptions of the most typical manifestations of the disease including fever, cough, diarrhea, fatigue, and severe respiratory distress, more recent papers addressed the involvement of the CNS by the virus aggression and on the neurological manifestations in affected patients [9].

The virus can cross the blood–brain barrier (BBB) and infect the neurons and glial cells that express ACE2, and this induce neuroinflammation and neuropathogenesis in the brain regions including the hypothalamus, which controls various physiological functions like hormonal balance [10,11]. The virus seems to enter the brain by passing the BBB through the nasopharyngeal epithelium via the olfactory nerve [12]. However, it may also reach the cerebral vasculature through the general circulation breaching the blood–brain barrier and invading and injuring the brain structures binding to its receptor ACE2 expressed in endothelial cells of cerebral capillaries and within the brain parenchyma in both neurons and microglia [13].

While, on the one hand, this can cause multiple hematological events such as thrombocytopenia, coagulopathy, and platelet dysfunction [14], it may also cause cerebrovascular disorders, including ischemic stroke, cerebral venous thrombosis, and cerebral hemorrhage that may be added to those regarding the cardio-respiratory system in both elderly and adult patients, especially if affected by hypertension or other cardiovascular risk factors [15]. SARS-CoV-2 can alter the hypothalamus–pituitary axis through several mechanisms [16].

The main pathological conditions of this axis so far reported as triggered by SARS-CoV-2 are pituitary apoplexy, hyponatremia and hypophysitis [12]. A study on post-mortem histopathology of pituitary and adrenal glands of COVID-19 patients did not evidence specific COVID-19-related changes in these glands, and the finding of pituitary necrosis was interpreted as a shock reaction [17]. However, the detection of SARS-CoV virus in adrenal and pituitary glands of four patients who died from COVID confirmed that these organs could be targeted by this virus since their tissues do express ACE2 [13,16,18]. In fact, an autopsy study on pituitary tissues from patients who died following SARS-CoV-2 infection confirmed that ACE2 mRNA was expressed in hypothalamus and pituitary gland cells, and that these cells also express TMPRSS2 receptors [19,20,21]. Moreover, an autopsy hypothalamus specimen from a patient- who died from COVID-19 was found to express the SARS-CoV-2 genome [15,21,22].

A possible COVID-19 indirect effect on the hypothalamic–pituitary axis may be mediated by cytokines that can trigger hypothalamus–pituitary inflammation with consequent pituitary dysfunctions [15]. The hypothalamus–pituitary–adrenal (HPA) axis plays a key role in stress response evoked by systemic inflammation, in acute-phase reaction, and the defense response at the tissue level [23]. Indeed, activation of HPA with hypercortisolemia can reduce cytokine toxicity and autoimmunity. When the HPA axis is activated, corticotropin-releasing hormone (CRH) and arginine-vasopressin (AVP) are secreted from the hypothalamic paraventricular nucleus. CRH and AVP stimulate the anterior pituitary to secrete ACTH which, in turn, stimulates the production of cortisol from the adrenal cortex [24].

However, this physiological cascade of events is altered in patients infected by SARS-CoV-2, as the virus triggers more complex mechanisms that impair the chrono-organization of this axis. To this regard, recent studies showed that SARS-CoV-2 may deflect the host’s immune response by expressing an amino-acid sequence that mimics the human adrenocorticotropic hormone (ACTH).

This plan of action induces the production of autoantibodies against ACTH that prevent the suitable adrenal response to the stress.

In fact, when the host produces antibodies against the viral antigens, these antibodies bind to ACTH, thus hindering its action on the adrenals. This event limiting HPA activity and secretion of corticosteroids could lead to adrenal cortisol insufficiency [13,25] (Figure 1).

However, several studies reported that, in severely ill COVID-19 patients, hypercortisolemia is prevalent and could be the consequence of decreased secretion of CBG (cortisol-binding globulin) and associated reduced serum protein binding. High cortisol levels during critical illness are also caused by the reduced activation of the enzymes responsible for its metabolic degradation and by the extension of the half-life of circulating cortisol [23,26,27].

Das et al., in their study, measured ACTH levels and showed that they tended to be lower, even if not significantly, in patients with moderate to severe disease [28]. Therefore, there is an ACTH and cortisol dissociation due to the cortisol secretion disengaged from ACTH, but dependent on other factors such as cytokines, which seem to be highly expressed in the more severe cases [23]. In fact, an increase in inflammatory cytokines may be favoured by the interference of the virus action with the ACTH function. Thus, this cascade of events may be interrupted by an appropriate early glucocorticoid therapy [16,25].

## 4. Hypothalamic-Pituitary Autoimmunity in SARS-CoV-2 Infection

The first question is whether the virus of this pandemic can cause neuroendocrine abnormalities through an immune-mediated aggression to the hypothalamic–pituitary axis, as already reported in patients infected by SARS-CoV-1 [15,25].

In fact, Leow et al. had previously studied the functional disorders of hypothalamic–pituitary–adrenal axis in patients infected by SARS-CoV-1, a coronavirus that caused the severe acute respiratory syndrome (SARS) several years ago, by investigating any chronic endocrine sequelae in 61 patients who had survived this syndrome. The patients were analysed for hormonal imbalances three months after recovery and periodically thereafter. Basal adrenal function was studied in all patients; then, those with impaired cortisol levels were submitted to a low dose (1 μg) ACTH test. Basal cortisol levels <138 nmol/L and cortisol response to ACTH stimulus <550 nmol/L allowed the diagnosis of adrenal insufficiency. In total, 24 out of 61 patients (39.3%) had evidence of central hypocortisolism, the majority of which resolved within a year, and there was an association with central hypothyroidism and low dehydroepiandrosterone sulphate in 4 of them. Even if an immunological study was not performed in these patients, the authors concluded that their results highlighted a possible etiologic role of SARS-associated coronavirus in causing a reversible autoimmune hypophysitis, with the pituitary–adrenal (HPA) axis more frequently affected [18,29], or in causing a direct hypothalamic damage responsible for transient hypothalamic-pituitary dysfunction [13,16,29]. Regarding COVID-19 patients, it first had to be demonstrated whether SARS-CoV-2 could induce the same alterations capable of evoking hypothalamic–pituitary autoimmunity. Concerning this, Gonen et al., in collaboration with our group, investigated neuroendocrine changes, in particular secondary adrenal insufficiency, using a dynamic test and searching for antipituitary (APA) and antihypothalamus (AHA) antibodies in order to clarify the possible role of autoimmunity in pituitary dysfunction of COVID-19 patients. In total, 49 patients with COVID-19 and 28 healthy controls were examined. The frequency of adrenal insufficiency in patients with COVID-19 was found in 8.2%. Patients with COVID-19 also had lower free T3, IGF-1, and total testosterone levels, and higher prolactin (PRL) levels. Among the four patients found to have an adrenal insufficiency, three of them were found to be APA positive and one AHA positive. None of the controls indicated an APA- or AHA-positive result. This study concluded that COVID-19 may cause secondary or tertiary adrenal insufficiency and that pituitary/ hypothalamic autoimmunity may play a role in this dysfunction; thus, a routine screening of adrenal function, as well as AHA and APA evaluations in these patients, should be recommended [30,31,32]. Considering these results, even if studies on the pituitary involvement in acute phase of this infection are scarce, a sequela of pituitary dysfunction, especially linked to pituitary autoimmunity in patients hospitalized from the acute phase of this infectious disease, could not be excluded. In fact, the infundibular hypothalamic–pituitary structures, due to their peculiar anatomical and vascular characteristics, may be very vulnerable to necrotic, ischemic, thrombo-embolic and hypoxic changes evoked by CNS virus invasion. In addition, because hypothalamus and pituitary tissue do express ACE2, they can therefore be viral targets. Moreover, the proposed hypothesis regarding patients infected by SARS-CoV of a molecular mimicry between some amino acid sequences of this virus and some sequences of the host ACTH chain, could explain how the immune response to viral particles may inactivate the host circulating ACTH, blunting the stress-induced cortisol rise. Concerning this, edema and neuronal degeneration, along with SARS-CoV genome, have also been found in the hypothalamus-pituitary region at autopsies of these patients [25]. Previous studies reported that hypothalamic-pituitary autoimmunity may be triggered by several CNS diseases, particularly traumatic injury, vascular alterations and infectious diseases. Brain alterations related to these diseases and mediators of inflammatory process (cytokines, free radicals, amino acids, and nitric oxide), can promote the activation of the immune system through the acceleration of neuronal cell necrosis. This may allow the unmasking of sequestered antigens at pituitary or hypothalamic levels with consequential production of respective autoantibodies that could contribute to late hypothalamic-pituitary dysfunction in these patients [33]. Pituitary dysfunction associated with the presence of AHA and/or APA at high titre as previously described in patients after traumatic brain injury, Sheehan’s syndrome, and infectious CNS diseases, may all cause alterations of anatomical and vascular brain structures. In particular, with regard to infectious diseases, Tanrivedi et al. performed an interesting study that investigated the role of autoimmune mechanisms in the pathogenesis of acute meningitis-induced hypopituitarism, searching prospectively for APA and AHA in 16 affected patients in acute phase, and at 6 and 12 months after recovery from acute meningitis. A single pituitary hormone deficiency was diagnosed in 18.7% of patients in acute phase, whereas at 12 months, 6 patients had isolated or had multiple pituitary hormone deficiencies. The occurrence of AHA and APA positivity was substantially high in these patients, ranging from 35 to 50%, and suggesting a possible role of autoimmunity in the pathogenesis of pituitary dysfunction, also after acute infectious meningitis [31]. It may be hypothesized that SARS-CoV-2 may induce the same alterations capable of evoking hypothalamic-pituitary autoimmunity, as suggested by the results of the study by Gonen et al. [30]. Thus, a prolonged follow-up of patients who survived acute COVID 19 infection, after hospital discharge, should be advisable to search for the presence of antihypothalamus and antipituitary antibodies, and to investigate the dynamic hormonal secretions of pituitary and satellite glands with appropriate tests. This could allow for a diagnosis of the possible occurrence of autoimmune pituitary dysfunction in the subclinical phase, such as a deficiency of HPA axis secretions and central hypothyroidism, hypogonadotropic hypogonadism, growth hormone deficiency and diabetes insipidus. This is particularly important since clinicians were initially advised against the use of corticosteroid therapy in COVID-19 patients because it may favour the cellular diffusion of this virus and reduce the immune response to the virus aggression. Instead, an early corticosteroid therapy, at the first appearance of isolated impairment of HPA axis secretions, could avoid the possible negative effects of secondary or tertiary adrenal insufficiency in these patients as well as the progression to more complex pituitary dysfunction by interrupting the autoimmune process.

Another important point is that SARS-CoV-2 infection may induce immune response hyperactivity involving Th1/Th17 lymphocytes and release of pro-inflammatory cytokines (IL-1β, IL-8, IL-10, IL-17, IFNγ, TNFα), which may cause a “cytokine storm” [34].

The “cytokine storm”, with interleukin-6 elevation, causes inflammatory thyroiditis [34,35], disruption of deiodinases and thyroid hormone transport proteins, and impaired TSH secretion, with consequent alteration of thyroid functional parameters. Indeed, correlated with the increase in IL-6, there is a decrease in free T3 concentration [36], due to a reduction in deiodinase activity causing decreased conversion of T4 to T3. A previous study showed that patients who died from COVID-19 had lower FT3 on admission compared with the survivors [37], and the decreases in FT3 levels were positively correlated with the severity of the disease [38].

Hypothalamus involvement in the SARS-CoV-2 aggression to central nervous system may also cause central hypogonadism, considering that physiologically hypothalamic GnRH regulates the secretions of the pituitary–gonadal axis [8].

The impaired activation of immune cells like lymphocytes and macrophages due to COVID-19 leads to a high release of inflammatory cytokines and oxidative stress in organs such as the brain and testes [39,40]. Proinflammatory cytokines and oxidative stress are highly harmful to steroidogenesis and spermatogenesis in the testes, and also compromise the fertility of affected patients [41,42].

## 5. Anti-Pituitary and Anti-Hypothalamus Antibodies Detection

Hypothalamic/pituitary autoimmunity may also be revealed by the presence of anti- hypothalamous (AHA) and anti-pituitary (APA) in sera of affected patients in analogy with other organ-specific antibodies whose detection helps to diagnose their respective autoimmune endocrine diseases [43]. However, due to various problems on the methods of detection of such antibodies and on the still-discussed clinical interpretation of the results, the role of APA and AHA in hypothalamic-pituitary autoimmunity is still debated [44,45], and because the true antigens reacting with these antibodies are still unknown [46,47,48].

Several antigens have been proposed as responsible for the autoimmune hypophysitis (LYH) and as a target of APA. In particular, pituitary GH1 and placental GH2 [49], alpha-enolase [50,51], pituitary gland specific factors 1a and 2 [49], secretogranin II [52], CG199 and somatomammotropin [53], PGSF1a, PGSF2, neuron-specific enolase (NSE), and corticotroph-specific transcription factor (TPIT) [48] have been proposed as autoantigens, but their pathogenic role is still a topic of discussion.

Several methods have been suggested for the detection of APA and AHA with the immunofluorescence and immunoblotting methods being the most widely employed. Using the immunofluorescence method, APA and AHA are detected on the cryostat section of the pituitary and hypothalamus obtained from humans or animals (even if the influence of human or animal pituitary substrate on the results of the immunofluorescence method is still discussed) and processed with fluorescein isothiocyanate conjugated goat anti-human Ig sera.

By this method, APA/AHA will react with cytoplasmic antigens, as distinct from pituitary hormones [54]. The pituitary/hypothalamic hormone-secreting cells targeted by these antibodies may be identified using a four-layer double fluorochrome immunofluorescence method. By this method, sera positive for APA/AHA are retested in a second step against the same pituitary/hypothalamic section and the animal’s pituitary/hypothalamic antiserum. The different colour of the anti-Ig conjugate against human and the anima serum, respectively green (FITC) and red (rhodamine), allows direct visual assessment of whether the patient’s serum and the animal anti-hormone serum stain the same or different pituitary/hypothalamic cells, thus allowing identification of the kind of pituitary/hypothalamic hormone-secreting cells targeted by these antibodies [54,55].

Using these methods, Bottazzo et al. demonstrated that APA exclusively recognized prolactin (PRL)-secreting cells [54,55,56]. However, none of the patients who were positive for PRL-cell antibodies showed an impairment of the pituitary function. Subsequently, APA selectively staining growth hormone (GH) pituitary cells were demonstrated in a girl with Turner’s syndrome and partial GH deficiency (GHD) [55], and in a patient with idiopathic GHD [56]. A further problem with immunofluorescence was that APA was not only found in some patients with biopsy-proven LYH [57,58] or in patients with suspected LYH, but also in patients with non-autoimmune pituitary diseases such as pituitary adenomas or empty sella syndrome [59,60]. The immunoblotting method [61] utilizes a homogenate of human autopsy pituitary tissue as a substrate to identify the antigen target of APA. By this method, Crock [50] showed that serum antibodies against a 49 kDa pituitary cytosolic protein were present in 70% of patients with biopsy-proven LYH. In 55% of patients with suspected LYH, including patients with isolated ACTH deficiency, patients with hypopituitarism associated to other autoimmune diseases or females with Sheehan’s syndrome [50]. Subsequently, these authors identified the 49 kDa pituitary cytoplasmatic protein as an alpha-enolase, which is an enzyme ubiquitously expressed and considered the antibodies to this antigen as a marker of LYH [62]. Subsequent studies have argued that antibodies to alpha-enolase cannot be considered specific for LYH because they are frequently present not only in patients with LYH, but also in some patients with hypopituitarism secondary to pituitary adenomas or to other pituitary diseases [51]. The previous arguments have led to attempts to improve the sensitivity and specificity of the immunoflurescence method. To this purpose, in our lab, we considered reliable results of APA and AHA detected by this method, only for titres starting from a determinate cut-off onwards and when the immunofluorescence pattern involved some, but not all, of the pituitary or hypothalamic hormone-secreting cells. This procedure allowed us to improve sensitivity and specificity of the immunofluorescence method also when employing as substrate pituitary and hypothalamic specimens from young baboons.

In 2012, in cooperation with the components of the Italian Autoimmune Hypophysitis Network Group, we performed a study involving 95 APA-positive patients affected by LYH, by searching for AHA in their sera to verify whether a possible involvement of hypothalamus autoimmunity could contribute to causing pituitary hormone alterations in these patients [63]. AHA and APA were detected by immunofluorescence method. Subsequently, the positive sera were retested by four-layer double immunofluorescence to ascertain the hypothalamic/pituitary hormone-secreting cells targeted by these antibodies.

As a result, both arginine-vasopressin-secreting cells (AVPc) and releasing hormone-secreting cells were targeted by AHA.

In particular, the detection of AHA targeting CRH-secreting cells in patients with GH/ACTH deficiency, but with APA specifically directed only to GH-secreting cells, suggested that some pluritropinic deficiencies in some patients with LYH may be due to combined autoimmune aggression both at pituitary and hypothalamic level [63]. In conclusion, even if the immunofluorescence method is widely employed, particularly due to the use of different human or animal substrates, the current results appearing in the literature are sometimes contradictory. However, we believe that the exclusion of low titres and confounding immunostaining patterns could improve the specificity and the sensitivity of the method getting trusted results for diagnosing, or at least suspecting, hypothalamic-pituitary autoimmunity while also using animal substrates, especially when the results are validated by using a second step with the four-layer double immunofluorescence.

Thus, we believe that, searching for APA and AHA in post-COVID patients who have presented virus invasion at cerebral level may help to identify those with possible hypothalamic-pituitary autoimmunity.

## 6. May the Autoimmune/Inflammatory Syndrome Induced by Adjuvants (ASIA) of the SARS-CoV-2 Vaccine Evoke Neuro-Immune-Endocrine Disorders?

The introduction of the SARS-CoV-2 vaccine was important in reducing the severe effects of the pandemic. However, recent reports highlighted the importance of adjuvants in the mRNA vaccine in inducing possible adverse effects on the endocrine glands through alteration of the autoimmune system, even if some aspects have yet to be clarified.

In particular, autoimmune thyroid dysfunctions and type 1 diabetes mellitus have been reported as possible events evoked by the autoimmune/inflammatory syndrome induced by adjuvants (ASIA) [64,65,66,67,68] but, to our knowledge, studies on hypothalamic-pituitary autoimmunity following this vaccination are lacking. However, a recent paper reported a case of isolated ACTH deficiency following immunization with SARS-CoV-2 vaccine [69]. A healthy 31-year-old man, one day after the second injection, noticed general fatigue and fever, and some days later he developed headaches, nausea, and diarrhea.

Laboratory findings showed hyponatremia, hypoglycemia and extremely low plasma ACTH and cortisol levels.

He was emergently treated with hydrocortisone. Magnetic resonance images revealed an atrophic pituitary gland [68]. Even if a study of hypothalamic-pituitary autoimmunity were not performed in this case, the occurrence of a possible autoimmune hypophysitis causing secondary hypodrenalism cannot be excluded, also considering that adrenocorticotrophs are usually among the first cells aggressed by the pituitary autoimmune process. In this connection, to search for anti-hypothalamus and anti-pituitary antibodies in patients presenting with these symptoms after vaccination should be advisable in order to confirm the autoimmune process and to possibly therapeutically interrupt the cascade of events leading to stable secondary hypoadrenalism.

## 7. Conclusions

SARS-CoV-2 infection may induce endocrine disorders in acute phase and in post- COVID period, particularly due to the effects of this virus at CNS level involving the hypothalamic-pituitary axis as well as a direct aggression to the peripheral glands. The aggression to hypothalamus-pituitary region may also elicit an autoimmune process involving this axis, which may be revealed by searching for APA/AHA in sera of affected patients. The consequent hypothalamic/pituitary dysfunction is responsible for functional disorders of the satellite glands as adrenal, thyroid and gonadal dysfunctions, in addition to pituitary alterations involving GH and prolactin secretions that have so far been reported. Thus, we believe that searching for APA and AHA in post-COVID patients who have presented virus invasion at cerebral levels may help to identify those with possible autoimmune pituitary–hypothalamic dysfunction in an early stage, thus preventing more severe endocrine disorders.

However, the extent to which COVID-19 contributes to short- and long-term effects of infection to the endocrine system is being currently discussed and remains an active area deserving more detailed research.

## Figures and Tables

**Figure 1 biomedicines-10-02855-f001:**
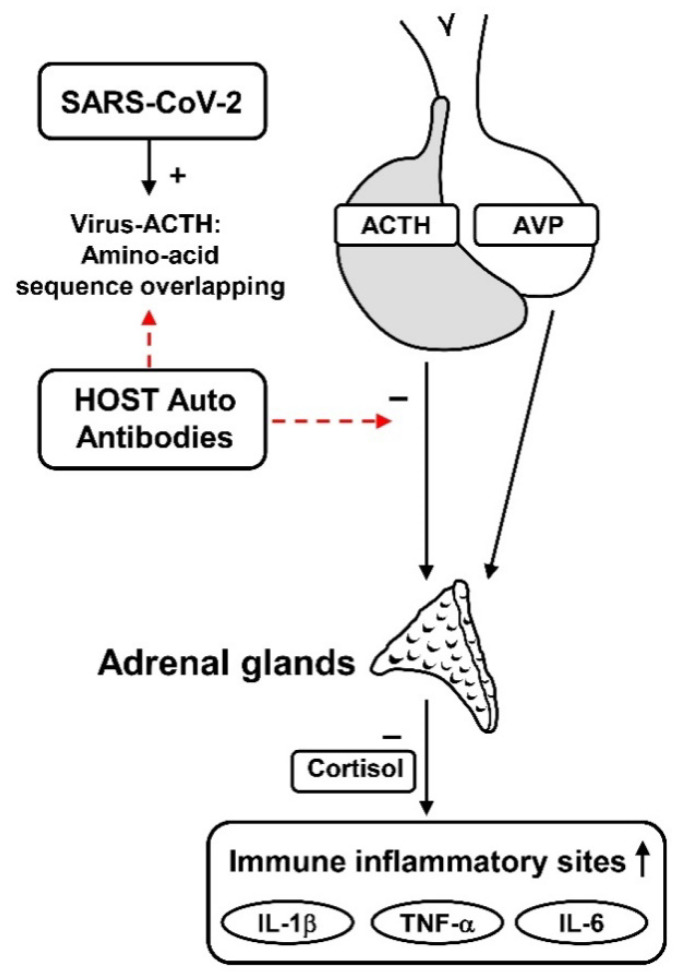
Possible strategy by which SARS-CoV-2 may deflect the host’s immune response. The amino-acid sequence, which the virus shares with the ACTH, induces the production of autoantibodies against this pituitary hormone that prevents the suitable adrenal response to the stress. ACTH: adrenocorticotropic hormone; AVP: arginine-vasopressin; IL-1β: interleukin 1 beta; IL-6: interlukin 6; TNF-α: Tumor Necrosis Factor α; +: increase; −: negative effect; ↑: Increase.

## Data Availability

Not applicable.

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
