# Peer review of "Neuroimmunoendocrinology of SARS-CoV-2 Infection"

_biomedicines, 2022, doi:10.3390/biomedicines10112855_

Round 1

Reviewer 1 Report

Dear Author,

This is an interesting paper based on a good core idea.

Here are my observations/questions/comments:

1.    Title  - Please correct “SARS-CoV” (capital letter for “V”)

2.    Abstract – Instead of “paying particular attention”, please use standard terminology as “we included as methods … types of studies/papers…”

3.    Abstract and Conclusion of the main text - “Adrenal, thyroid and gonadal dysfunctions, but also GH and prolactin alterations have been so far reported” Please do not mix glands with hormones.

4.    Abstract – Conclusion – I suggest use your own take home message rather than “further…”

5.    Introduction – Please rephrase this “Dysregulation of endocrine functions is related to different disorders including adrenal dysfunction, hypothyroidism, hypogonadism, anxiety, stress, and depression”.

I suggest “Endocrine dysfunctions are reported in …” or something clear.

6.    Chapter 2 – The  hypothalamus-pituitary axis is also affected as a response to adrenal or thyroid or even gonadal damage (feedback, not direct lesion)

7.    Figure 1. Please explain all the nomenclature – IL, IFN, etc.

8.    Tittle of the Chapter 3 is not clear since an autoimmunity – based condition concerning hypothalamus-pituitary axis is also an endocrine disorder

9.    Chapter 4 – The title is too long. Is it a question?

10. The Methods of research should have a special section. In 10 years from now, the reader will need to calculate the years of research and which publications were studied

11. A subsection of Discussions should include the strength and the limits of this topic and your our ideas in terms of further exploration of this subject and/or even potential new concepts that may be released (if any).

Best regards,

Author Response

We are very indebted to this Reviewer for his comments and suggestions. The enclosed manuscript has been revised along the lines indicated. We are pleased to reply

Reviewer 1

Dear Author,

This is an interesting paper based on a good core idea.

Here are my observations/questions/comments:

1.Title  - Please correct “SARS-CoV” (capital letter for “V”).

This has been corrected

  1. Abstract – Instead of “paying particular attention”, please use standard terminology as “we included as methods … types of studies/papers…”

      Abstract has been modified as suggested

  1. Abstract and Conclusion of the main text - “Adrenal, thyroid and gonadal dysfunctions, but also GH and prolactin alterations have been so far , reported” Please do not mix glands with hormones.

      The sentence has been modified as follows: Adrenal, thyroid and gonadal dysfunctions, but also pituitary alterations,  involving particularly GH and prolactin secretions, have been so far reported.

  1. Abstract – Conclusion – I suggest use your own take home message rather than “further…”

      Abstract and Conclusions have been modified as suggested.

  1. Introduction- Please rephrase this “Dysregulation of endocrine functions is related to different disorders including adrenal dysfunction, hypothyroidism, hypogonadism, anxiety, stress, and dep ression”.

      I suggest “Endocrine dysfunctions are reported in …” or something clear.

       This has been rephrased as suggested: Endocrine dysfunctions are reported in several gland secretions involving adrenal, thyroid, gonads, associated with anxiety, stress, and depression.

  1. Chapter 2 – The  hypothalamus-pituitary axis is also affected as a response to adrenal or thyroid or even gonadal damage (feedback, not direct lesion)

ACE mRNA expression has been demonstrated in hypothalamus and pituitary gland cells and SARS-CoV virus has been detected at hypothalamic-pituitary level in patients died for this infection, thus suggesting a possible direct aggression to this axis with consequent primary impairment of its hormone secretions and not only as feedback alteration

  1. Figure 1. Please explain all the nomenclature – IL, IFN, etc.

     This has been made

  1. Title of the Chapter 3 is not clear since an autoimmunity – based condition concerning hypothalamus-pituitary axis is also an endocrine disorder

       Following the comment of this Reviewer, the title of chapter 3 (now chapter 4) has been modified by deleting “and related endocrine disorders”

  1. Chapter 4 – The title is too long. Is it a question?

      The title of this chapter (now number 5) has been modified as follows: Anti-pituitary and anti-hypothalamus antibodies detection

  1. The Methods of research should have a special section. In 10 years from now, the reader will need to calculate the years of research and which publications were studied

      We added a special section.

  1. A subsection of Discussions should include the strength and the limits of this topic and your our ideas in terms of further exploration of this subject and/or even potential new concepts that may be released (if any).

      We think that in Conclusions the strength and the limits of this topic  were sufficiently specified as well as the necessity of further studies to better clarify the aspects of this topic

Reviewer 2 Report

Dear Authors,

Very interesting and breath-taking manuscript, reading of what took from me very short time. Thank you for this inside story of SARS and neuroimmunoendocrine system!

However, I have some small remarks/advice to improve a little bot your exciting manuscript: 1) you have indicated the data search only in the abstract, but I would like to ask you to add this also in short paragraph at the end of 1st section (including the time period for the issued literature search (from-to), indicating the key words, and exclusion criteria); 2) References are OK, but as you have used also 12 previous century sources (and they are good and fit very well into the context of manuscript), I would like to ask you to change the search of literature where you indicate "...so far" with mentioning also additionally "some historical aspects" of the topic. Otherwise your main story is about the last years and 1980-1990ies actualities what rises the  additional questions and contradict each with other. 3) some small grammar mistakes, - the lost points at the end of sentences also should be corrected.

Author Response

We are very indebted to this Reviewer for his comments and suggestions. The enclosed manuscript has been revised along the lines indicated.

Dear Authors,

Very interesting and breath-taking manuscript, reading of what took from me very short time. Thank you for this inside story of SARS and neuroimmunoendocrine system!

However, I have some small remarks/advice to improve a little bot your exciting manuscript: 1) you have indicated the data search only in the abstract, but I would like to ask you to add this also in short paragraph at the end of 1st section (including the time period for the issued literature search (from-to), indicating the key words, and exclusion criteria); 2) References are OK, but as you have used also 12 previous century sources (and they are good and fit very well into the context of manuscript), I would like to ask you to change the search of literature where you indicate "...so far" with mentioning also additionally "some historical aspects" of the topic. Otherwise your main story is about the last years and 1980-1990ies actualities what rises the  additional questions and contradict each with other. 3) some small grammar mistakes, - the lost points at the end of sentences also should be corrected.

Points 1-3 of these Reviewer’s comments have been accurately considered in revising our paper. A special sections about Methods was added.